# Learning Speech Representations from Raw Audio by Joint Audiovisual Self-Supervision

**Abhinav Shukla** [1]   **Stavros Petridis** [1 2]   **Maja Pantic** [1 3]

## Abstract

The intuitive interaction between the audio and visual modalities is valuable for cross-modal self-supervised learning. This concept has been demonstrated for generic audiovisual tasks like video action recognition and acoustic scene classification. However, self-supervision remains under-explored for audiovisual speech. We propose a method to learn self-supervised speech representations from the raw audio waveform. We train a raw audio encoder by combining audio-only self-supervision (by predicting informative audio attributes) with visual self-supervision (by generating talking faces from audio). The visual pretext task drives the audio representations to capture information related to lip movements. This enriches the audio encoder with visual information and the encoder can be used for evaluation without the visual modality. Our method attains competitive performance with respect to existing self-supervised audio features on established isolated word classification benchmarks, and significantly outperforms other methods at learning from fewer labels. Notably, our method also outperforms fully supervised training, thus providing a strong initialization for speech related tasks. Our results demonstrate the potential of multimodal self-supervision in audiovisual speech for learning good audio representations.

## 1. Introduction

Self-supervised learning of representations from large unlabeled datasets is a popular contemporary trend in machine learning. After being widely adopted in areas like natural language processing and computer vision, self-supervision is now rapidly developing as a noteworthy topic in audio and speech processing. Self-supervision aims to capture the most informative properties from the underlying structure of unlabeled data to learn generalized representations. This is extremely promising in problem settings involving a large amount of unlabeled data but limited labeled data. In the context of audio and speech processing, this is relevant to low resource languages, emotion recognition, cross-cultural speech recognition and other such problems with small-sized datasets. Even though there has been recent research interest in self-supervised learning for speech data, most works focus only on the audio modality alone. Audiovisual speech data offers interesting possibilities for cross-modal self-supervision, which is something relatively lesser explored. In this work, we present a method for self-supervised representation learning of audio features that leverages both the audio and visual modalities. We demonstrate how generating a talking lip video from a single frame and the corresponding audio can be used as a pretext task for visual self-supervision to train a raw audio encoder. We combine this with audio-only self-supervision based on predicting informative audio attributes, similar to (Pascual et al., 2019). This results in an audio encoder trained by joint audiovisual self-supervision. We evaluate the method on spoken word classification and achieve competitive results when comparing with existing self-supervised methods. Our method also results in significantly better performance when learning with limited data (10 % of training set) for the downstream tasks. Importantly, our method also outperforms fully supervised training (directly training the encoder on the downstream task). Our observations motivate the utility of self-supervised pretraining for audio related tasks. We demonstrate that cross-modal supervision in audiovisual speech can learn better representations compared to unimodal audio-only or visual-only self-supervision.

### 1.1. Related work

Self-supervised learning has been very influential in recent advances in natural language processing (BERT (Devlin et al., 2018), RoBERTa (Liu et al., 2019) etc.) and computer vision (CPC (Oord et al., 2018), MoCo (He et al., 2020), PIRL (Misra & van der Maaten, 2019) etc.). It is

Abhinav Shukla's work was supported by a PhD scholarship by Samsung Electronics, UK. [1]Imperial College London, UK [2]Samsung AI Centre, Cambridge, UK [3]Facebook London, UK. Correspondence to: Abhinav Shukla <a.shukla@imperial.ac.uk>.

*Published at the workshop on Self-supervision in Audio and Speech at the $37^{th}$ International Conference on Machine Learning*, Vienna, Austria. Copyright 2020 by the author(s).

also beginning to mature as a relevant topic in audio and speech processing. CPC (Contrast Predictive Coding) (Oord et al., 2018) was a seminal work in self-supervised learning which also demonstrated the applicability of contrastive self-supervised learning to audio. Wav2vec (Schneider et al., 2019) refines the idea from CPC specifically for speech. CPC based self-supervision has also been shown to generalize well to multiple languages (Rivière et al., 2020). APC (Autoregressive Predictive Coding) (Chung et al., 2019) is a similar approach that predicts the next token of a speech segment from the history. Another very relevant recent work is PASE (Problem Agnostic Speech Encoder) (Pascual et al., 2019), which aims to learn multi-task speech representations from raw audio by predicting a number of handcrafted features such as MFCCs, prosody and waveform. Teacher-student models have also been explored for audio self-supervision where the trained model from a previous epoch acts as the teacher model for the next epoch (Kumar & Ithapu, 2020). All of the works discussed so far are unimodal audio-only self-supervised methods. There are also a few other works that utilize both audio and visual information. There are multiple ways to capture this cross-modal interaction including audiovisual synchronization (Owens et al., 2018), cross-modal transition modeling (Pham et al., 2019), cross-modal pseudolabel based clustering (Alwassel et al., 2019), contrastive learning (Tian et al., 2019; Patrick et al., 2020), and audiovisual instance discrimination (Morgado et al., 2020). However most of these works present cross-modal self-supervision in the context of generic audiovisual data, with application to tasks like video action recognition and acoustic scene classification. There is limited work that explores self-supervision specifically in the context of audiovisual speech. We have explored this concept in recent related work (Shukla et al., 2020c;b;a). This work extends the idea from our prior work. Specifically, we move from learning speech representations directly from raw audio instead of from mel features. We also adopt a different and more refined approach for audio-only self-supervision (described in Section 2.3).

## 2. Method

### 2.1. Audio encoder architecture

We use a 1D Resnet18 (He et al., 2016) encoder as the backbone for all of our proposed methods (detailed architecture in appendix). The encoder $f_a$ (see Fig. 2 and 3) takes as input a 16 kHz raw audio waveform and converts it into a 512-D audio feature vector for every timestep. The output sample rate is 25 audio feature vectors per second, which matches that of 25 FPS video in the LRW dataset. This allows us to have a one-to-one mapping between the two modalities, which helps in cross-modal learning and allows us to avoid oversampling or undersampling either modal-

ity. Other contemporary self-supervised methods (Alwassel et al., 2019; Patrick et al., 2020) use a 2D Resnet18 audio encoder operating on mel features (operating similar to image based CNNs). However, we wanted our audio encoder to directly operate on the raw audio waveform and perform end-to-end self-supervised representation learning without starting from an intermediate feature like MFCCs or log mel spectrograms, which is why we chose a 1D Resnet18.

### 2.2. Visual Self-Supervision

For visual self-supervision, we generate a talking lip video from a still image and the corresponding audio (see Fig. 1 and Fig. 2). The model is comprised of three components: (i) the audio encoder $f_a$ (1D Resnet18), (ii) the identity encoder $f_{id}$, and (iii) the frame decoder $f_d$. The model operates on 1 second long segments from an audiovisual speech dataset. The audio encoder $f_a$ (Fig. 2 bottom-left) converts the 1 second audio sample $x$ into a 512 dimensional embedding with 25 timesteps ($z_{aud}$). The identity encoder $f_{id}$ (Fig. 2 top-left) is a 6 layer CNN that converts the mouth region of the first video frame $x_{im}$ (a 64x64 image) into a 64 dimensional identity embedding ($z_{id}$). This embedding is replicated 25 times to match the timesteps of the audio embedding. The latent representation $z$ is the concatenation of $z_{aud}$ and $z_{id}$ (as shown in Fig. 2). This then goes through the frame decoder $f_d$ (see Fig. 2 top-right), which is a CNN that uses strided transposed convolutions to generate the video frames of the lip movements. The skip connections between the identity encoder and frame decoder help in preserving subject identity in the generated frames. An L1 reconstruction loss between frames from the generated video ($f_d(z)$) and those from the real video ($y_{video}$) is used to train the network. We use the L1 loss as opposed to the L2 loss to get relatively sharper reconstructions. Our model aims to predict lip movements given only audio and speaker identity information from the first frame. In this process, the audio encoder is driven to produce useful **speech features that correlate with lip movements** (because accurate lip movement reconstruction will reduce the loss). The audio features obtained by reconstructing lip movements are likely to contain information about the speech content. Our proposed method is related to our prior work on visual self-supervision to learn audio features (Shukla et al., 2020c;b;a). In this work, the key difference is that we use a raw audio encoder for end-to-end learning as opposed to the log mel spectrogram encoder we used in (Shukla et al., 2020b;a). Also, instead of reconstructing the full face, we focus on the mouth region which contains visual information about the speech content, which we hypothesized would lead to better representations for speech recognition.

$$z(x, x_{im}) = cat(f_a(x), f_{id}(x_{im})) \qquad (1)$$

$$L_{video}(x, x_{im}) = |f_d(z(x, x_{im})) - y_{video}| \qquad (2)$$

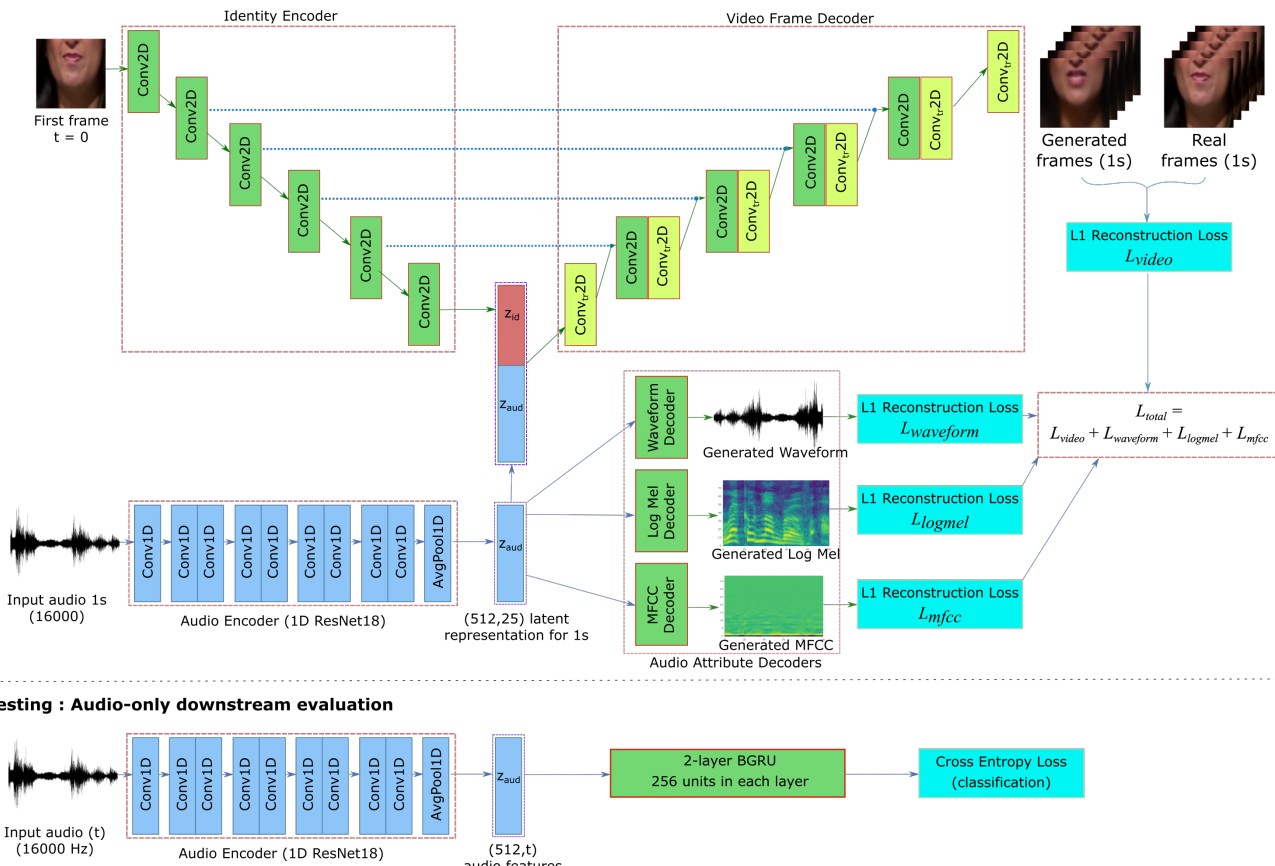

*Figure 1.* An illustration of the encoder-decoder model we use for joint audiovisual self-supervision. From an unlabeled sample of audiovisual speech, we use the raw audio waveform and the first video frame to generate a talking lip video. Lip movement reconstruction offers visual self-supervision. We also use decoders to reconstruct salient audio attributes (MFCCs, log mel, waveform) for audio-only self-supervision. By jointly optimizing the reconstruction losses for both modalities, we get joint audiovisual self-supervision. The trained audio encoder can then be used for audio-only downstream tasks.

*Table 1.* Results for spoken word classification (Accuracy in %) on the Speech Commands (SPC, 30 classes) (Warden, 2018) and the Lip Reading in the Wild (LRW, 500 classes) (Chung & Zisserman, 2016) datasets. For evaluation, a 2 layer GRU model is used on the encoder outputs for each pretraining method, before finetuning on the downstream task.

| Pretraining method | Self-supervision | Input type | Dataset and % of Labels used | | | |
|---|---|---|---|---|---|---|
| | | | SPC 100% | SPC 10% | LRW 100% | LRW 10% |
| MFCC | - | - | 94.33 | 87.08 | 90.16 | 37.56 |
| PASE (Pascual et al., 2019) | Audio | Raw audio | 95.61 | 83.81 | 93.40 | 1.88 |
| APC (Chung et al., 2019) | Audio | Mel features | 94.87 | 89.91 | 93.97 | 57.41 |
| wav2vec (Schneider et al., 2019) | Audio | Raw audio | 96.04 | 91.57 | 94.60 | 19.50 |
| L1 (Shukla et al., 2020b) | Visual | Mel features | 95.11 | 86.43 | 94.45 | 33.43 |
| L1 + Odd (Shukla et al., 2020b) | Audiovisual | Mel features | 95.77 | 90.16 | 94.72 | 67.98 |
| Ours (A) | Audio | Raw audio | 95.06 | 90.56 | 94.14 | 69.70 |
| Ours (V) | Visual | Raw audio | 94.38 | 88.31 | 92.18 | 52.99 |
| Ours (AV) | Audiovisual | Raw audio | 95.21 | 90.63 | 95.37 | 77.13 |
| Supervised 1D Resnet18 | - | Raw audio | 93.79 | 81.12 | 90.34 | 13.72 |

## 2.3. Audio Self-Supervision

In prior work (Shukla et al., 2020b), we employed temporal order based pretext task for audio-only self-supervision (predicting which of the inputs are jumbled or reversed). We wanted to examine whether it is possible to yield better speech representations using a more refined pretext task. In this work, our methodology for audio-only self-supervision is inspired from PASE (Pascual et al., 2019). We predict three informative audio attributes: (i) MFCCs, (ii) Log mel spectrograms, and (iii) the waveform. The key difference of our method with PASE is the fact that we directly train a 1D Resnet18 encoder model on the raw audio waveform. PASE requires intermediate steps like adding speech distortions for data augmentation, SincNet filters, and a penultimate Quasi-RNN layer. We also adopt only 3 of the most informative predicted attributes from PASE for simplicity. Fig. 3 illustrates our method for audio-only self-supervision. The audio encoder ($f_a$) converts 1 second of 16 kHz input audio ($x$) into a 512 dimensional audio embedding ($z_{aud}$) with 25 timesteps (exactly the same as in the method for visual self-supervision). The audio representation is then used as input to three separate decoders ($f_{mfcc}$, $f_{logmel}$ & $f_{wav}$) that reconstruct the desired audio attributes. We keep the decoder architectures as simple as possible in order to incentivize the important information about the audio attributes to be captured by the audio encoder. The MFCC and the log mel spectrogram decoders (Fig. 3 right) are both comprised of a single fully connected layer of 256 units. The waveform decoder (Fig. 3 top-left) is made of a transposed convolution layer followed by a convolution layer that outputs the reconstructed waveform (in an autoencoder-like fashion). We use an L1 loss between each reconstructed attribute with its ground truth ($y_{attrib}$) to train the model. The total loss is the sum of the MFCC loss, the log mel loss, and the waveform loss. For $attrib \in \{mfcc, logmel, wav\}$, the loss is:

$$L_{audio}(x) = \sum_{attrib} |f_{attrib}(f_a(x)) - y_{attrib}| \quad (3)$$

## 2.4. Audiovisual Self-Supervision

For joint audiovisual self-supervision (see Fig. 1), we simply combine the two proposed methods for visual-only and audio-only self-supervision. Since the same audio encoder architecture has been used in both models, we can simply use the shared audio representation as input to each of the four decoders (frame decoder, MFCC decoder, log mel decoder, waveform decoder). The total loss is the sum of the audio-only and the visual-only losses. The audio encoder ($f_a$) is thus trained end-to-end and is driven to produce features that contain information about each of the predicted attributes from both the audio and the visual modalities.

$$L_{total}(x, x_{im}) = L_{video}(x, x_{im}) + L_{audio}(x) \quad (4)$$

## 3. Experiments

**Datasets** The LRW dataset (Chung & Zisserman, 2016) is a large, in-the-wild dataset of 500 different isolated words primarily from BBC recordings. It is an audiovisual speech dataset and is thus appropriate for training our methods. We use a subset of LRW that has only nearly frontal videos (with yaw, pitch and roll restricted to a maximum of 10 degrees), in order to have a cleaner supervisory signal from the visual modality. This filtering leaves us with a total of around 40 hours of usable data. We use this subset of the LRW dataset for self-supervised pretraining of our proposed methods. We also use it as a spoken word classification evaluation dataset. The SPC (Speech Commands v0.01) dataset (Warden, 2018) contains 64,727 total utterances of 30 different words by 1,881 speakers. We use SPC also as a spoken word classification evaluation dataset.

**Baselines** We compare our methods against other self-supervised methods for learning speech representations. For all the baselines, we use the code (and pretrained models) provided by the authors. We compare against PASE (Pascual et al., 2019), APC (Chung et al., 2019) and wav2vec (Schneider et al., 2019). We also compare against our prior related work. L1 (Shukla et al., 2020b) is similar to our proposed method for visual-only self-supervision but is based on log mel spectrograms as opposed to raw audio. L1 + Odd (Shukla et al., 2020b) is an audio-visual self-supervised method. We use a more refined audio self-supervision approach in this work. We also compare our methods against two supervised learning baselines for audio. We use 39 dimensional MFCCs (13 coefficients, 13 deltas, and 13 delta-deltas) as the first supervised baseline. The second baseline is a fully supervised 1D Resnet18 model (same architecture as our pretrained encoders but trained from scratch directly on the evaluation datasets).

**Experimental setup** We evaluate all methods on isolated word classification on the Speech Commands (SPC) (Warden, 2018) and Lip Reading in the Wild (LRW) (Chung & Zisserman, 2016) datasets. We use a 2 layer BiGRU (with 256 units in each layer) on the encoder outputs followed by a linear layer with as many units as the number of target classes (30 for SPC, 500 for LRW). This acts as the downstream classifier and remains the same for every method. For downstream classification, we finetune the models (as shown in bottom of Fig. 1) for 50 epochs. The learning rate is 0.0001 for the first 40 epochs and 0.00001 for the last 10 epochs. We use the standard softmax + cross entropy loss for training. We opted to use a BiGRU for simplicity, however this can be replaced by any model that can classify variable length sequences into discrete categories (such as LSTMs, TCNs, LiGRUs (Ravanelli et al., 2018)). The results can be seen in Table 1.

*Table 2.* Results for spoken word classification (Accuracy in %) under various levels of introduced noise (SNR in dB). Babble noise from the NOISEX database is used to perturb the audio samples in the LRW and SPC datasets.

| Dataset | Model | Noise level (SNR) | | | | | | |
|---------|-------|-------|------|------|-------|-------|-------|-------|
| | | -5 dB | 0 dB | 5dB | 10 dB | 15 dB | 20 dB | Clean |
| SPC | MFCC | 76.31 | 84.97 | 90.56 | 91.98 | 93.05 | 94.19 | 94.33 |
| | Ours (A) | 79.35 | 88.42 | 92.34 | 93.41 | 94.63 | 95.04 | 95.06 |
| | Ours (V) | 77.92 | 86.92 | 91.01 | 92.80 | 93.47 | 93.88 | 94.38 |
| | Ours (AV) | 79.79 | 88.69 | 92.21 | 93.57 | 94.65 | 95.02 | 95.21 |
| LRW | MFCC | 50.18 | 70.75 | 81.08 | 85.74 | 88.41 | 90.11 | 90.16 |
| | Ours (A) | 58.84 | 79.13 | 89.14 | 91.72 | 92.87 | 93.84 | 94.14 |
| | Ours (V) | 51.40 | 73.47 | 84.61 | 88.11 | 90.98 | 91.58 | 92.18 |
| | Ours (AV) | 64.63 | 82.59 | 90.08 | 92.09 | 92.91 | 93.87 | 95.37 |

**Results with all labels** With 100% of the training dataset used, all self-supervised methods achieve comparable performance and outperform fully supervised training. On the SPC dataset, the best overall performance is attained by wav2vec with an accuracy of 96.04%, followed by our prior work at 95.77%, PASE at 95.61% and our proposed method at 95.21%. On LRW, the best performance is by our method with an accuracy of 95.37%.

**Learning with fewer labels** The concept of self-supervision is especially relevant to situations where labeled data is scarce. To compare the methods in such situations, we perform the same word classification experiments on the SPC and LRW datasets but with only 10% of the samples being used in the training set (the validation and test sets remain unchanged). Note that we completely omit the remaining 90% of the training set (see Tables 6, 7, 8 for exact split details). This leaves us with around 170 training examples per class for the SPC dataset (30 classes) and only around 20 training examples per class for the LRW dataset (500 classes). This makes the problem significantly more challenging. On SPC, there is a slight degradation in the performance of all methods. Our method attains an accuracy of 90.63% which is second to only wav2vec at an accuracy of 91.57%. On LRW, all other methods get severely affected and overfit to the small training set. Our method is the least affected and significantly outperforms all other methods with a best performance of 77.13%.

**Noisy situations** We also compare the performance of the variations of our method under various levels of artificially induced noise. We introduce babble noise from the NOISEX (Varga & Steeneken, 1993) database to create noisy versions of the SPC and LRW datasets. We use six levels of noise, in the range of -5 dB SNR to 20 dB SNR in increments of 5 dB. The results for the noisy datasets can be seen in Table 2. All our methods outperform MFCCs at all noise levels on both datasets. The joint audiovisual method is the best.

## 4. Discussion

There are multiple interesting observations from our obtained results. Audio-only supervision yields better results than visual-only supervision. However, the model trained with joint audiovisual self-supervision performs better than the models trained with unimodal audio-only and visual-only self-supervision in almost all scenarios. including noisy datasets. This highlights the utility of the complementary information encoded by visual self-supervision and demonstrates the potential of multimodal self-supervision as a useful tool in speech representation learning. Also notably, despite all tested methods being very similar in performance on the full datasets, there is a clear gap when using a small training set and our method is the best at learning with fewer labels, which is very relevant to low resource domains. This can have significant impact in problems like low resource language ASR, emotion recognition and cross-cultural ASR. Our method also significantly outperforms fully supervised training from scratch, which further motivates the utility of self-supervised pretraining for speech.

**Future work** This is a work in progress and there are many other speech related applications that we can evaluate our model on. In this work, we only focused on the classification of isolated words. We will also test the model on continuous CTC based speech recognition on datasets like Librispeech and TIMIT, and other tasks like speaker identification and speech emotion recognition. An especially relevant application would be low resource language ASR. There are also interesting directions to explore to improve our method. In this work, we exhibit how joint audiovisual information can be used for audio representation learning. In a similar manner, we could also utilize this cross-modal information for visual representation learning (e.g. predicting speech attributes from the visual modality). Another interesting line of work is multimodal contrastive self-supervised learning which has been demonstrated for generic audiovisual data but not for audiovisual speech.

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

# Appendix

## A. Audio encoders

*Table 3.* Encoder type and number of trainable parameters in each of the compared methods.

| Method | Encoder type | Parameters |
|---|---|---|
| PASE | SincNet + CNN + FC | 5,818,020 |
| APC | Log mel + GRU | 4,105,296 |
| wav2vec | CNN | 32,537,088 |
| L1 + Odd | Log mel + GRU | 4,065,282 |
| Ours | 1D Resnet18 | 3,848,576 |

*Table 4.* Feature dimensionality and sample rate of each of the compared methods.

| Method | Dim. | Hz |
|---|---|---|
| MFCC | 39 | 101 |
| PASE | 100 | 100 |
| APC | 512 | 101 |
| wav2vec | 512 | 98 |
| L1 | 512 | 101 |
| L1 + Odd | 512 | 101 |
| Ours | 512 | 25 |

*Table 5.* Pretraining dataset and duration for each method

| Method | Pretraining Dataset | Duration |
|---|---|---|
| PASE | Librispeech subset | 10 hours |
| APC | Librispeech train-clean-360 | 360 hours |
| wav2vec | Full Librispeech + WSJ | 1000 hours |
| L1 | LRW frontal subset | 36 hours |
| L1 + Odd | LRW frontal subset | 36 hours |
| Ours | LRW frontal subset | 36 hours |

**Pretraining datasets for baselines** The results in Table 1 for all the baseline methods (PASE, APC, wav2vec) have been computed using the public code and pretrained models provided by the authors. These baseline methods (and our method) have been pretrained on varying amounts and types of data. For a completely fair comparison, all methods need to be pretrained with the same data. We experimented with pretraining all baseline methods on the same 36 hour LRW frontal subset that we use for our method. The results obtained with the baseline methods using this approach were either equivalent or worse to those with the public pretrained models. This shows that our model may be able to learn better representations on the same amount of pretraining data. However for the results, we use the public pretrained models which may assist with reproducibility.

## B. Dataset and split details

*Table 6.* The number of data samples in each split of each dataset.

| Dataset - % labels | Split size (samples) | | |
|---|---|---|---|
| | Train | Val | Test |
| SPC-100% | 51088 | 6798 | 6835 |
| SPC-10% | 5097 | 6798 | 6835 |
| LRW-100% | 112812 | 5878 | 5987 |
| LRW-10% | 11054 | 5878 | 5987 |

*Table 7.* The duration (in hours) of each split of each dataset.

| Dataset - % labels | Split duration (hours) | | |
|---|---|---|---|
| | Train | Val | Test |
| SPC-100% | 14.19 | 1.89 | 1.90 |
| SPC-10% | 1.41 | 1.89 | 1.90 |
| LRW-100% | 36.35 | 1.89 | 1.92 |
| LRW-10% | 3.56 | 1.89 | 1.92 |

*Table 8.* The average number of samples (rounded to nearest integer) and duration in minutes of each class in the training set.

| Dataset | Classes | n/class | t/class |
|---|---|---|---|
| SPC-100% | 30 | 1703 | 28.38 |
| SPC-10% | 30 | 170 | 2.82 |
| LRW-100% | 500 | 225 | 4.36 |
| LRW-10% | 500 | 22 | 0.42 |

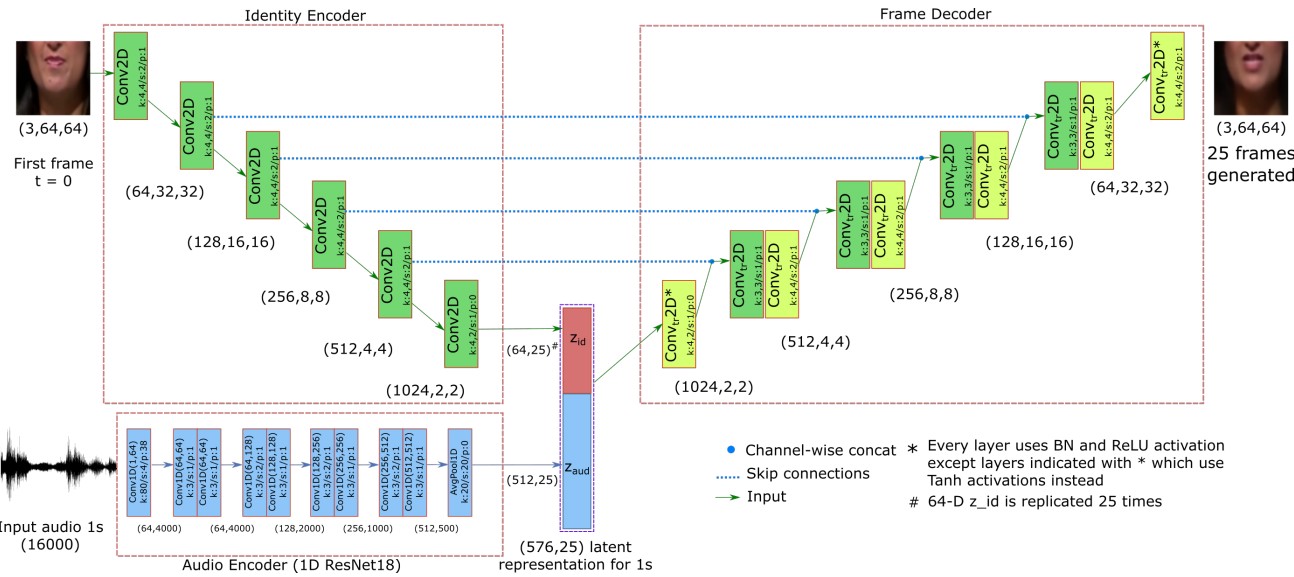

*Figure 2.* A detailed illustration of the encoder-decoder model we use for lip video reconstruction. From an unlabeled sample of audiovisual speech, we use the audio and the first frame of the video ($t = 0$) to generate a video with $t$ frames. The model contains: (1) an identity encoder which produces a 64-D identity embedding; (2) an audio encoder which converts the input audio (t frames of 80 dimensional log mel spectrograms) into a 512-D audio embedding; (3) a frame decoder which generates video from the concatenated latent representation using transposed convolutions.

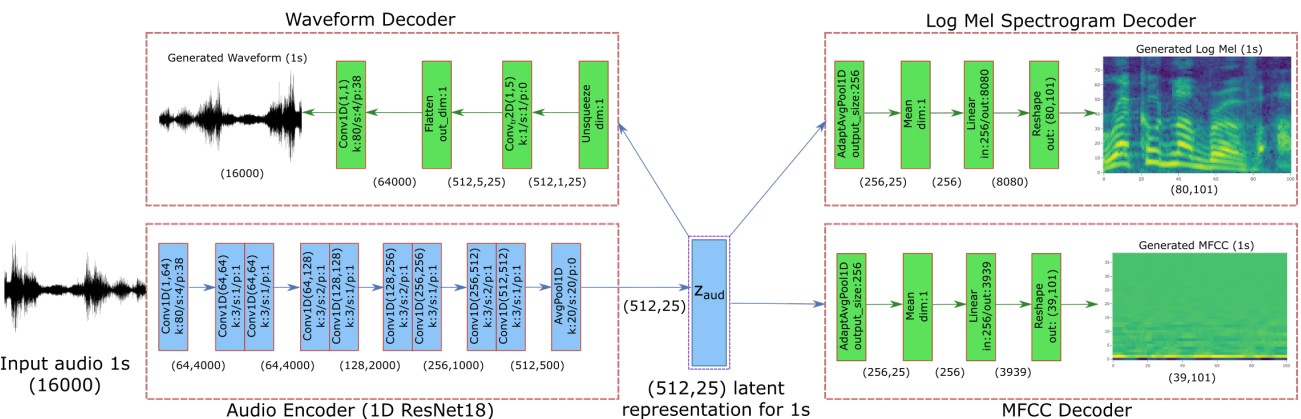

*Figure 3.* A detailed illustration of the encoder-decoder model we use for audio-only self-supervised representation learning. From an input waveform of 1 second, we predict three informative attributes: MFCC, log mel spectrogram and the waveform. The decoders are kept as simple as possible to incentivize the audio representations to capture the necessary information about the attributes.