# OpenReview forum: "Learning Speech Representations from Raw Audio by Joint Audiovisual Self-Supervision"
_ICML.cc/2020/Workshop/SAS — SAS 2020_

### Official Review · AnonReviewer3 · 2020-06-28
**Joint audiovisual embeddings for speech**

**Rating:** 7
**Confidence:** 3

**Review:**

Summary
The authors propose an approach to learning representations of speech multimodally, by incorporating a visual self-supervised task to synthesize video of mouth movements along with audio self-supervision.  Thorough experiments using two downstream tasks (word classification and lip reading) show comparisons to other types of learned speech representations including the authors' own prior work and other popular systems / features, and show especially strong results with large improvements over completely supervised learning when only a small amount of labeled data is available.

Pros
* Very thorough experiments comparing to other systems, and also evaluating performance in a noisy setting, on two different downstream tasks.
* Strong results over fully supervised learning when smaller quantities of labeled data are available.
* use of published datasets is good for reproducibility.

Cons
* It's difficult to assess the novelty of this work.  It sounds like it's very similar to the authors' previous work, except that they are using raw waveforms instead of Mel, and reconstructing mouth movements only instead of video of the entire face.  It is also stated that "a different and more refined approach for the audio self-supervision component" is used, but section 2.3 about audio self-supervision only describes differences to PASE and not to their own prior work.  It would help to more clearly state what's new in this paper in relation to the previous work, and why these differences are being explored.
* It would be helpful to the reader to include a sentence or two of analysis to explain what can be concluded or taken away from the "noisy situations" experiments (table 2).
* a minor point: in the section "Results with all labels," it is stated that "all methods [...] outperform fully supervised training."  This doesn't appear to be strictly true -- the "LRW 100%" results for MFCC are slightly worse than the fully-supervised case.

---

### Official Review · AnonReviewer2 · 2020-06-29
**A well written paper and interesting results, albeit with some missing information**

**Confidence:** 4
**Rating:** 6

**Review:**

The paper evaluates the idea of using visual information about lip-movements to learn a better, i.e. more discriminative, encoder for audio in a self-supervised manner.

The abstract, together with Section 1, gives a good motivation of the core idea and is very well written. Section 2 describes the main contributions, i.e. the method and architecture. Here, Figure 1 includes some very small and hard-to-read fonts.

The novelty of this section and of the paper overall seems a bit limited: As the authors write, the main difference to their current work at ICASSP2020 is their choice of features - audio samples instead of log mel spectra, and output image segments - mouth regions instead of whole faces.

In Section 3, the experiments are introduced and explained. The downstream task that is considered in the evaluations is isolated word recognition. In the given context, I can understand that this may be of interest, and it does not, in my view, call the results into question. That being said, I believe that works aiming for a better identifiability of spoken words should be able to take the sequential and continuous nature of speech signals into account, so the approach of simplifying speech recognition to isolated word classification, which is bound to fail in any realistic scenario, does mark a small weakness of the approach.

In the evaluation, there is a bit more information and one comparison missing:
- why is there no comparison between the classification based on audiovisual self-supervision and a classifier based on audio-visual features?
- also, why are there no confidence regions associated to the results and why are no significance tests performed to assess the significance of differences? How would the entire test results differ with a different random initialization of all involved models?
- finally, why does the evaluation in noise only consider the own approach? One may hypothesize that mel spectra could be a bit more resilient to noise, so it would be nice to verify (and possibly also reject) this hypothesis, especially with the proposed contribution being centered around using raw audio data, with its potential vulnerability under such noisy conditions.

The paper concludes with a discussion in Section 4. I am generally interested in the planned future steps, but I am not sure how the future suggested work on TIMIT and Librispeech will be addressable with the proposed framework, which requires, to my understanding, the availability of video data.

The paper is generally very well written and the following are only minor points:
- Hyphenation is a bit inconsistent. Not all compound adjectives or adverbs are hyphenated.
- There is an incomplete sentence starting with: Specifically, we move from using visual self-supervision learn speech features directly

---

### Official Review · AnonReviewer1 · 2020-06-30
**Self supervised speech representations learning employing Residual Neural Network by leveraging Audio-Visual Modalities**

**Confidence:** 5
**Rating:** 6

**Review:**

In this work, authors presented 1-D ResNet-based self-supervised speech representations learning framework that leverages both audio and visual modalities.  This paper is well- organized and well-written to some extent and reported good performances over the baseline.
(1) Self-supervised approach provided better results than the fully supervised approach. Perhaps authors could provide some insights why this self-supervised approach performed better than the fully supervised approach.
(2) Some rewriting/modification is needed in table 1 i.e., in the field headings (encoder or encoding? and so on).
(3) With MFCC-based first supervised approach which classifier did you use (table 1, MFCC)?
(4) Self-supervision is a form of unsupervised learning where supervision is provided by the data and it provides an opportunity to leverage a large amount of unlabelled data with a small amount of labelled data while learning is carried out in a supervised manner.
I have some queries about the results reported In table 2 (with 100% and 10% labels in the training data).
(a) When 100% labels of all training data are used then all the approaches are working in a fully supervised fashion?
(b) When 10% of the labels in the training data are used (in table 1): In this case self-supervised approaches are using 100% of training data but considering only 10% as labelled data and the rest 90% as unlabelled data, am i correct?  And, for fully supervised approaches you are using only 10% of the labelled training data? I think author should made it clear in section 3 of the paper.

In terms of Quality of References: Misses some relevant works.
In terms of Clarity of presentation: Clear enough but could benefit from some revision.
In terms of Originality: As mentioned by the authors this work is an extension of their previous work.  It seems original to me.
In terms of significance: Self supervised learning is an interesting area of research and it provides us an opportunity to utilize large unlabelled training with a small amount of labelled training data.

---

### Decision · Program_Chairs · 2020-07-01

**Decision:**

Accept

**Comment:**

Dear author(s),

Thank you very much for your submission at the ICML2020@SaS workshop (https://icml-sas.gitlab.io/). Based on the scores assigned by the reviewers, we are happy to notify you that your paper was accepted for the workshop.

Please, address the comments of the reviewers and submit the camera-ready version by July 8. We ask the authors to record a 15min video for your talk. At the workshop, we will have the pre-recorded video as well as a live QA session. It is important to keep this time limit, otherwise, your talk will be automatically cut. The deadline for uploading the video is July 8. The detailed instructions for uploading will follow.

Feel free to contact us for any questions!

Best,

The ICML20@SaS organizers:
Mirco Ravanelli
Titouan Parcollet
Dmitriy Serdyuk
Devon Hjelm
Bhuvana Ramabhadran